# Archaeometric Study of the White Marbles from "Madonna Della Febbre" Altar in San Domenico Church (Cosenza, Southern Italy)

**Natalia Rovella [1],\*, Stefania Bosco [2] and Donatella Barca [2]**

1   Department of Biological, Geological and Environmental Sciences (BIGeA), University of Bologna, via Sant'Alberto, 163, 48123 Ravenna, Italy
2   Department of Biology, Ecology and Earth Sciences (DiBEST), Ponte Bucci, University of Calabria, 87036 Arcavacata di Rende, Italy; stefania.bosco@unical.it (S.B.); donatella.barca@unical.it (D.B.)
\*   Correspondence: natalia.rovella@unibo.it

**Abstract:** San Domenico Church was built between 1441 and 1468 and represents one of the most important historical buildings of the Cosenza area (Calabria, Southern Italy) thanks to its architectonic style and the works inside, such as the "Madonna della Febbre", a notable marble altar dated back to the XVI century. The church, as well as the sculptural group, underwent various interventions over time, unfortunately scarcely documented; thus, in this paper, the characterization of six white marble samples coming from the altar, was carried out to determine their provenance. The samples were analyzed by means of complementary methodologies well known in the archaeometric field: polarized optical microscopy (POM); an electron probe micro analyzer coupled with an energy dispersive spectrometer (EPMA-EDS); inductively coupled plasma–mass spectrometry (ICP-MS); and isotope ratio mass spectrometry (IRMS). The results provided important information about the "Madonna della Febbre" altar, suggesting the presence of different typologies of marbles and hypothesizing their possible provenance, including Carrara and Docimium. It was not clear if these marbles were re-used materials but, regardless, the reported information adds precious elements to the history of the entire architectonic complex, providing new issues to be deepened.

**Keywords:** provenance; isotopic analysis; manganese content; petrographic characterization; Calabria



## 1. Introduction

The marble complex "Madonna della Febbre" is one of the most precious sculptures saved in San Domenico Church in Cosenza (Calabria, Southern Italy). Unfortunately, the historical information about the complex and the church, such as the provenance of materials, the architect, and the origin of the raw materials, is scarce and often unclear.

The church devoted to San Domenico was built between 1441 and 1468, and represents one of the most notable historical buildings of the Cosenza area [1]. It was financed by the Prince of Bisignano Antonio Sanseverino and, since 1525 has been a "*Studium Generale*", an important venue of the Dominican Monastic Order in Calabria that hosted the famous philosopher Tommaso Campanella in 1588.

San Domenico Church represents a unique example of the passage from the Gothic to the early Renaissance architectonic style in the area, and is one of only few in Southern Italy. It was realized by highly qualified workers, as the level of execution of many structural and decorative elements suggests. The church has a Baroque superstructure associated with the original Gothic one, with a barrel-vaulted dome, grafted onto the transept. The bell tower dates to the 17th century. The entire monumental complex has been gradually enriched over time by interventions that have expanded its structure, although unfortunately they have not always been documented.

The "Madonna della Febbre" altar most probably dates back to the 16th century [2], and it is held in the chapel of San Matteo [3]. The marble group traditionally was attributed to Giovanni da Nola [4] but, more recently, to two Spanish sculptors Bartolomé Ordoñez and Diego de Siloe [5]. Unfortunately, the available information is lacking as there is no data linking the various authors to the probable choice of a particular type of marble.

Specifically, the complex is located on the altar in the so-called "Rosariello" Chapel (because it allows access to the oratory of the Rosary), and it is constituted by whitish marble blocks without particular foliations. The altar (Figure 1) consists of a central aedicule hosting the Virgin with the Baby Jesus, supported by the right arm, and framed by double pilasters, trabeation, and frieze decorated with grotesque motifs; a lunette, bearing the figure of the bearded and blessing God the Father, stands above.

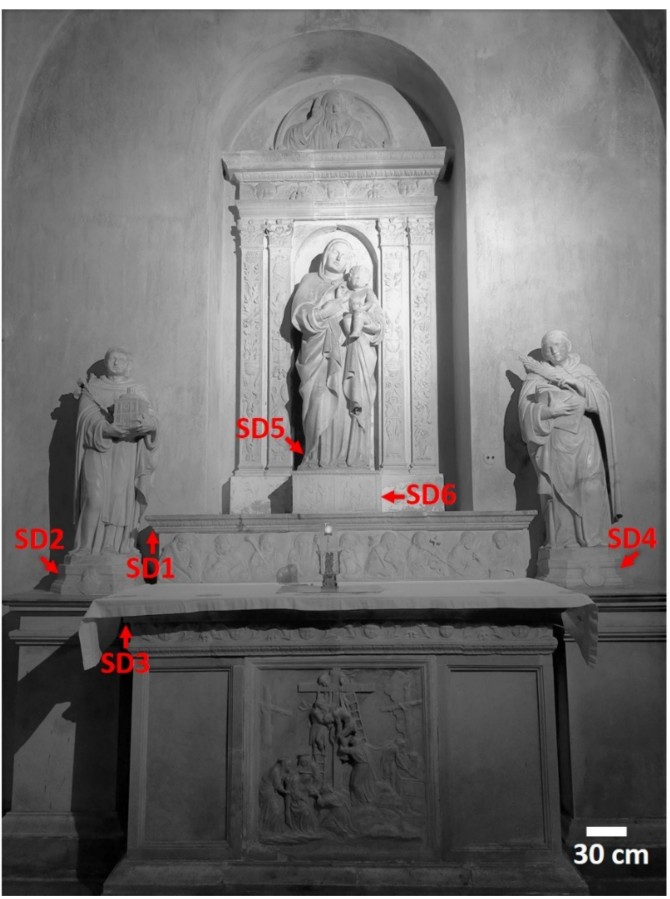

**Figure 1.** The marble complex "Madonna della Febbre" and details of the sample points.

The pedestal of the Madonna is adorned with the scenes of the Annunciation and the Resurrection in the two panels on the left and right, respectively, and with the Birth of Christ and the Adoration of the Magi in the center.

Two Dominican saints flank the Madonna: San Domenico on the left side and San Tommaso d'Aquino on the right side. The frontal shows, in bas-relief, a Deposition of Christ from the Cross.

Currently, the location of the dossal is not the original one; probably, subsequent interventions have led to different access points to the room with the consequence of the altar position being changed.

Moreover, the retable, because of these tamperings, has lost some portions, which were then partly found by Alfonso Frangipane in 1933. He identified the aedicules holding the statues of the Madonna and Dominican Saints [6]. The restructuring works of the 1940s caused the partial disappearance of the predella, where only eight Apostles are now visible on the sides of Christ.

Nowadays, there are no clear records about past interventions or restorations, nor the provenance of the marbles used for the sculptural group. In this regard, there are several source areas of this stone in the Mediterranean basin dislocated in different countries such as Italy, Greece, Spain, or Turkey [7]. The marbles have been amply used since the Classical Age, or in more recent times for architectural or decorative aims. Many studies focus on determining the provenance of these materials, highlighting also the complexity of this topic. This is linked to the microstructural and compositional heterogeneity of the materials, their widespread circulation and sale in all the Mediterranean basin, and their heavy reuse [7–18].

Concerning the marbles of San Domenico Church, the historical information about the exploitation sites of marble used in Cosenza are too fragmentary. The authors in [5] cite some sources about the intention of some collaborators of the Cosenza Curia, during those years, to go to the Carrara areas to choose some marble blocks, but no other trace of this trade was found in the urban archives.

Thus, the purpose of this work was to reconstruct the history of the complex of the "Madonna della Febbre" using characterizations of the marble constituting the sculpture. For this aim, an archaeometric approach based on complementary analytical methodologies was applied.

In this paper, samples of marble taken from the "Madonna della Febbre" underwent a petrographic and geochemical study in order to determine their provenance by means of POM (polarized optical microscopy (POM), electron probe micro analyzer coupled with an energy dispersive spectrometer (EPMA-EDS), isotope ratio mass spectrometry (IRMS) of $^{13}$C and $^{18}$O, and inductively coupled plasma–mass spectrometry (ICP-MS).

## 2. Materials and Methods

Six fragments of marble (~1 cm sized) were taken from the "Madonna della Febbre" altar under the supervision of an officer of Superintendence of Cosenza. The samples and their location are listed in Table 1 and shown in Figure 1. The sampling was carried out by scalpel in areas of the different marble blocks that were not visible, or from points that were already fractured so as not to further damage the sculpture; for this same reason, we were only allowed to take a few fragments of very small sizes.

**Table 1.** Marble samples taken from the "Madonna della Febbre" altar.

| ID Samples | Location |
|---|---|
| SD1 | Trabeation |
| SD2 | San Domenico (on the left) |
| SD3 | Frontal |
| SD4 | San Tommaso d'Aquino (on the right) |
| SD5 | Madonna |
| SD6 | Pedestal of Madonna |

The petrographic characterization was conducted by polarized optical microscopy (POM) on thin sections, using a Zeiss AxioLab microscope (Zeiss, Oberkochen, Germany), equipped with a digital camera Zeiss AxioCam MRc to acquire photomicrographs. The observations enable us to determine textural features such as the grain boundary shape (GBS), the shape preferred orientation (SPO) and the maximum grain size (MGS) of calcite and/or dolomite crystals [19].

The maximum grain size (MGS) of the calcite or dolomite individuals is an important diagnostic parameter, generally ascribable to the metamorphic grade reached by a marble during its genesis [20]. In the archaeometric research field, according to their grain size, marbles are conventionally divided into fine-grained (up to 2 mm), medium-grained (2–5 mm) and coarse-grained (above 5 mm) [21].

MGS was obtained by measuring (in mm) the larger grains in a sample; 6–7 measurements were acquired by Zeiss AxioCam MRc camera, and then the average value was considered for more representative data.

Electron probe microanalysis (EPMA) measurements were performed by a JEOL JXA-8230 (JEOL Ltd, Tokyo, Japan) using an accelerating voltage of 15 kV and an electric current of 10 nA. An energy dispersive X-ray analysis system (EDS) (JEOL Ltd, Tokyo, Japan) as coupled for microanalysis. A graphite coating was applied to make the surface of the thin section conductive for EDS analysis.

EPMA-EDS analyses were carried out to obtain the main composition of marbles (i.e., calcite or dolomite), and especially to identify the accessory minerals, crucial for the determination of the provenance [12,19].

These minerals are often too small and rare to be recognized by POM; on the contrary, EPMA-EDS provides more detailed data. In fact, it operates up to very high magnification (practically up to $10^4\times$, by contrast with common optical microscopes that operate up to 400–500$\times$ maximum), even the grains that are too small to be seen under an optical microscope can be detected and analyzed by electron microscopy [21].

Isotope ratio mass spectrometry (IRMS) was used to determine stable C and O isotope composition (i.e., $\delta^{13}C$ and $\delta^{18}O$). $CO_2$ from powdered samples was obtained by classical phosphoric acid–calcium carbonate reaction method by an automated carbonate preparation device (Thermo Scientific GasBench II) [22]. SD5 was not analyzed because of the insufficient amount of sample. The 44-mass peak area was used to determine the $CaCO_3$ content. The carbonate isotopic compositions ($\delta^{18}O$ and $\delta^{13}C$) were measured by Thermo Scientific Delta V Advantage continuous flow isotope ratio mass spectrometer (Thermo Fisher Scientific Waltham, MA USA). Results are expressed in delta ($\delta$) notation relative to the V-PDB standard. The precision of the carbon and oxygen isotope ratios for duplicate analyses were 0.1 and 0.2‰, respectively. Analyses were performed at the Department of Earth Sciences and Sea in University of Palermo.

Inductively coupled plasma–mass spectrometry (ICP-MS) was applied to determine the Mn content. Analyses were performed through a quadrupole Perkin Elmer/SCIEX ElanDRCe spectrometer (Perkin Elmer Inc., Waltham, MA, USA). The samples were broken into millimetric fragments and then powdered in an agate mortar. Using a Mars6 apparatus (CEM Corporation, Matthews, NC, USA) with Teflon (TFM) digestion vessels, 100 mg of powder was dissolved in a mixture of Merck "suprapur" quality acids, namely hydrofluoric acid (3 mL HF), nitric acid (5 mL $HNO_3$) and perchloric acid (6 mL $HClO_4$). Before complete evaporation of the acid, 2 mL of perchloric acid were added to ensure complete removal of the hydro-fluoric acid. To obtain the mother solutions, the reaction solutions were allowed to cool slowly and were diluted to 100 mL with Millipore water up to a sample/solution weight ratio of 1:1000. The same procedure was used to prepare one standard reference material: the micaschist (SDC-1) from the US Geological Survey. This was used as an unknown sample during the analytical sequence, as suggested by [23]. The concentration of Mn was determined via external calibration curves by using Merck "ICP multi-element standard solution VI". In addition, to compensate for potential errors linked, for instance, to possible drifts of the machine, a known amount of an internal standard (indium) was added to the standards and solutions. Three marble samples (SD2, SD3 and SD4) were analyzed, plus the quality control standard (Mica Schist SDC1 by USGS); the measurements were repeated three times, and the mean and standard deviation were calculated. Mn concentration reported in the text refers to the mean values. Moreover, three blank determinations, interspersed during the analytical run. It was not possible to analyze SD1, SD5, or SD6 because of the insufficient amount of available samples. To evaluate the accuracy, the mean values of measurements carried out on the quality control standard were compared with those certified. Accuracy was within ±5%. The instrument detection limit was evaluated by multiplying the standard deviation of the blanks by 3.

## 3. Results and Discussion

### 3.1. POM Analysis

The petrographic characterization was carried out considering the criteria individuated in literature such as [21,24], and described previously. The observations on thin sections allowed us to distinguish different typologies of marbles.

SD1 showed very different textural features than the other samples. It was characterized by a medium grain size (2 mm < MGS < 5 mm) [21] with an average MGS of 2.35 mm. The texture was heteroblastic, isotropic, with a vague mosaic fabric, and rare triple points. Many grains were strongly deformed as highlighted by twin lamellae of calcite crystals (Figure 2a). There was no SPO, crystal boundaries were not always well defined, and in some points they appeared from straight to embayed.

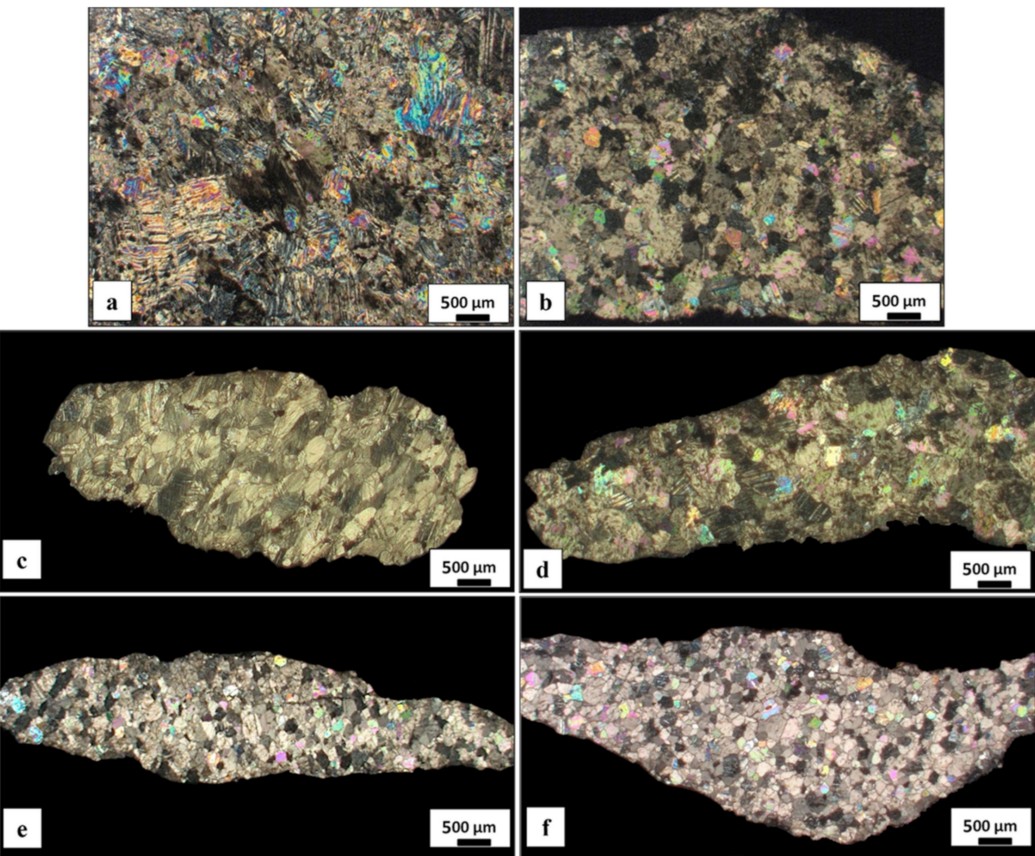

**Figure 2.** Microphotographs in POM (cross-polarized light view, Magnification 1×) showing the different textural features of the marbles constituting the "Madonna della Febbre" complex. (**a**) sample SD1; (**b**) sample SD2; (**c**) sample SD3; (**d**) sample SD4; (**e**) sample SD5; (**f**) sample SD6.

Petrographic features of SD2 and SD4 were similar (Figure 2b,d). The texture was heteroblastic, slightly lineated in some points, with fine grain size (MGS < 2 mm) and an average MGS of 0.60 mm.

The fabric was commonly of mosaic type, with occasional triple points and traces of lineation GBS varies from curved to embayed.

The texture in SD3 appeared heteroblastic, with a slight SPO and common strained crystals. The grain size was fine, with an average MGS of 0.81 mm. The fabric was mosaic-type (Figure 2c) with somewhat visible triple points; GBS varied from curve to embayed.

SD5 and SD6 showed a rather homeoblastic and isotropic texture with a very fine grain size being the average MGS < 0.5 mm. The fabric was mosaic and polygonal-type with clear triple points, whereas GBS was straight (Figure 2e,f).

Table 2 reports all the main petrographic features and the MGS values, crucial for the determination of the provenance of marble.

**Table 2.** Main mineralogical and petrographic features, carbon–oxygen isotopic data and the most probable provenance of the marble samples coming from the "Madonna della Febbre" altar. Legend: Ho. homeoblastic; He. heteroblastic; Is. isotropic; An. anisotropic; Ms. mosaic; SPO: Ob. observed; Nob. not observed; GBS: C. curved; E. embayed; S. straight. Minerals identified by EPMA-EDS: Cal. calcite as main mineral. Accessory phases: Ab. Albite; Ap. Apatite; Kfs. K-feldspar; Ms. Muscovite; Ox. Titanium oxide. Mn indicates the concentration of Manganese determined by ICP-MS.

| ID Samples | Texture | Fabric | SPO | GBS | MGS (mm) | Cal | Ab | Ap | Kfs | Ms | Ox | $\delta^{13}C$ (‰) | $\delta^{18}O$ (‰) | Mn (ppm) | Probable Provenance |
|---|---|---|---|---|---|---|---|---|---|---|---|---|---|---|---|
| SD1 | He; Is | Ms | Nob | S; E | 2.35 | x | | x | | | | 0.77 | −6.47 | - | Not Determined |
| SD2 | He; Is | Ms | Nob | C; E | 0.60 | x | x | | | | | 2.15 | −1.13 | 14.5 | Carrara? |
| SD3 | He; An | Ms | Ob | C; E | 0.81 | x | | x | | | | 0.83 | −6.42 | 379 | Docimium |
| SD4 | He; Is | Ms | Nob | C; E | 0.70 | x | x | | | | | 1.69 | −1.62 | 11.7 | Carrara? |
| SD5 | Ho; Is | Ms | Nob | S | 0.48 | x | x | x | | x | x | - | - | - | Carrara |
| SD6 | Ho; Is | Ms | Nob | S | 0.45 | x | x | x | x | x | x | 1.93 | −1.42 | - | Carrara |

Most of the samples (SD2, SD3, SD4, SD5, SD6) were very fine-grained marbles with MGS < 1 mm; only SD1 showed an MGS related to medium-grained marbles. Considering the box bars matching the samples, MGS provides various interpretations about the provenance as reported in the diagram in Figure 3: SD1 falls most likely in Aphrodisias, Proconnesos, Paros-2(3), Paros-4, Thasos-3 (i.e., Cape Vathy quarry in [20,25]), less probably, in Paros-1, Thasos-1(2) (i.e., Alikì and Cape Fanari quarries in [20,25], respectively), and Naxos districts. SD2, SD3 and SD4 likely originate from Carrara, Göktepe, Docimium (Afyon), Pentelicon or Hymettus. Finally, SD5 and SD6 fall into Carrara, Göktepe, Docimium (Afyon), Hymettus fields.

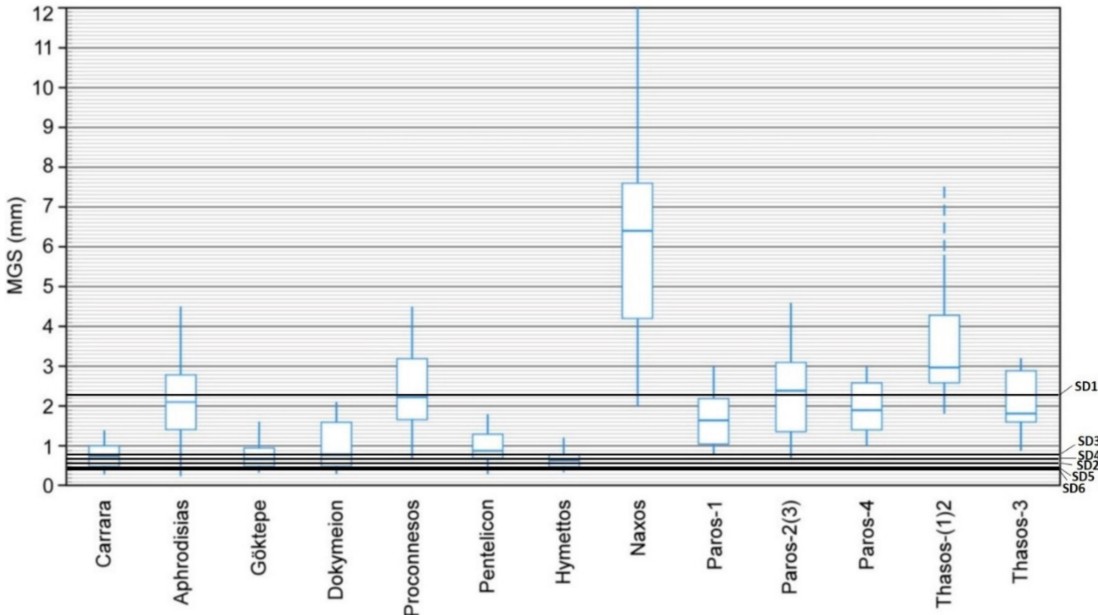

**Figure 3.** Maximum grain size (MGS) diagram modified from [21] related to the 10 chief white marbles used in classical times. (Notes: Dokymeion = Docimium (Latin version)/Afyon districts; Proconnesos = Marmara district; Hymettos = Hymettus district).

### 3.2. EPMA-EDS Analysis

Data suggests that calcite is the main mineralogical phase in all the samples. The accessory minerals are generally very tiny, ranging in size from 5 to 20 μm; the species

identified are reported in Table 2. Apatite is the most common one and it was individuated in SD1, SD3, SD5, SD6. Its presence is typical of numerous white marbles detected in the Mediterranean area [24], but generally such small crystals (<40 μm) are more abundant in the Carrara, Pentelicon, Paros, and Marmara districts [19].

Albitic plagioclase was found in SD2, SD4, SD5 and SD6, and it is common in Carrara white marble sources [21,24].

K-feldspar was present in SD6. Nevertheless, it can be found occasionally in archaeological marbles and, according to [24], its bearing on the provenance of host marbles is consequently unknown.

Tiny flakes of muscovite sized between 10 and 20 μm were observed in SD5 and SD6. The mineral is relatively frequent in many districts, e.g., Afyon/Docimium, Aphrodisias, Carrara, Paros, Pentelicon and Thasos [19,24].

Ti-oxides were identified in SD5 and SD6; they could be related mainly to Carrara, Pentelicon, Naxos, or Marmara, but they can also be found occasionally in the other marbles [19].

*3.3. Isotope Analysis and Mn Content*

IRMS analysis determined $\delta^{13}C$ and $\delta^{18}O$ isotopic ratios to further discriminate the possible provenance of the marbles. The values obtained are reported in Table 2, and in two diagrams edited by [21] related to fine-grained white marbles (Figure 4a) and medium/coarse-grained white marbles (Figure 4b), respectively. MGS data were used to distinguish the two typologies.

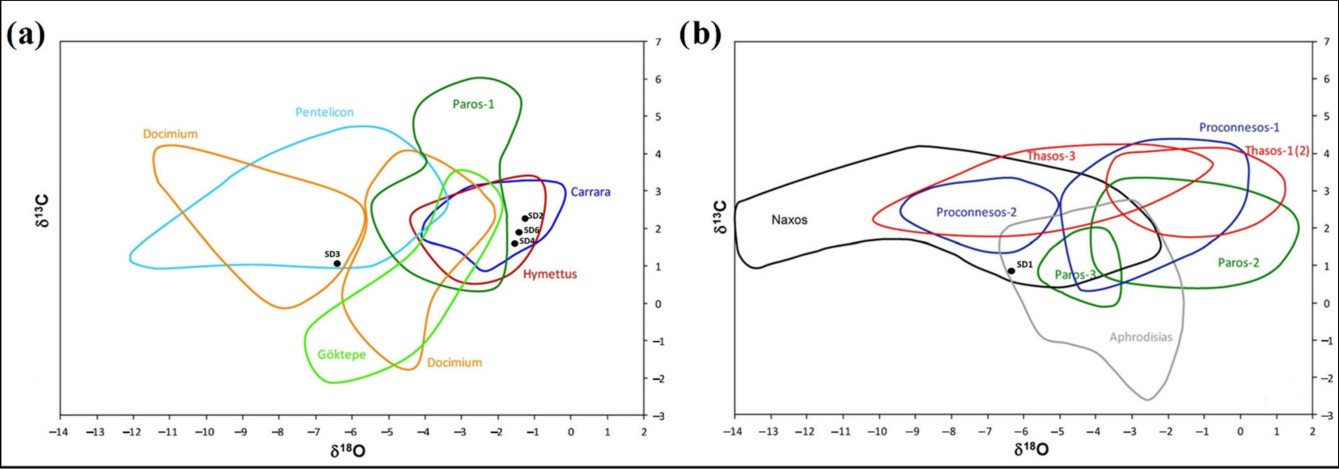

**Figure 4.** Comparison of the isotopic data about "Madonna della Febbre" samples and C and O stable isotope reference diagrams modified from [21] about the 10 most used Roman marbles for fine-grained (**a**) and medium to coarse (**b**) ones, respectively.

SD1 is the only sample to be plotted in Figure 4b, and falls in Aphrodisias and Naxos fields (Figure 4b). SD2, SD4 and SD6 match Carrara and Hymettus marbles, whereas SD3 is comparable with Docimium (Afyon) and Pentelicon districts. Generally, all the samples show overlaps in the different districts reported in the diagrams providing an unclear/partial interpretation of the provenance.

Considering the small amount of sample available, it was possible to determine the Mn only in SD2, SD3 and SD4 (Table 2 and Table S1). The results were plotted onto the diagram proposed by [20], where different marble districts are distinguished based on their Mn content (Figure 5). SD3 matches both the Pentelicon and Docimium (Afyon) quarries, and SD2 and SD4 fall in many fields (i.e., Carrara, Docimium (Afyon), Usak, Paros, Naxos, Thasos and Aphrodisias), making an unambiguous interpretation difficult and necessarily entailing a comparison with the other analytical techniques used.

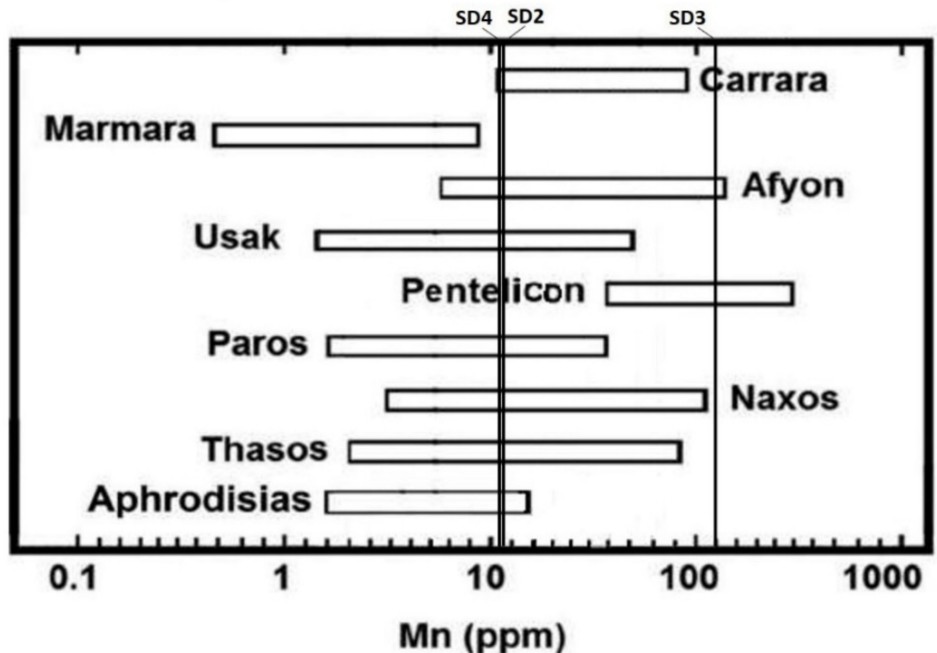

**Figure 5.** Diagram modified from [20] distinguishes the Mediterranean marbles based on the Mn content. Samples compositions were plotted, and their possible provenances are represented by the vertical lines crossing the different fields relative to marble districts. (Notes: Marmara = Proconnesos district; Afyon = Docimium district).

*3.4. Discussion*

The comparison between the data obtained from the different analytical methods enabled us to hypothesize the possible districts of provenance of the marbles used in the complex "Madonna della Febbre". All the marbles were calcite-based as reported in the EPMA-EDS analyses.

SD1 belonged to the medium-sized marbles. MGS measurements indicated Aphrodisias, Proconnesos, Naxos, Paros and Thasos districts, and overlayed isotopic ratios that suggested both Aphrodisias and Naxos as possible sources of SD1. Regarding Thasos, it is plausible to exclude Cape Vathy quarry as a dolomite marble [25], a mineral not revealed in this sample; moreover, although Alikì and Cape Fanari quarries consist of calcitic marble [25], their provenance is rather unlikely, as reported in MGS diagram. The petrographic observations were not clear enough to confirm a univocal hypothesis because the evident microstructural deformation undergone by the original rock made it difficult to recognize the main and diagnostic features of the marble, and consequently to discriminate the right source among those identified by IRMS and EPMA-EDS data. Of the latter marbles, small apatites <40 μm could suggest Paros provenance, but this thesis was not supported by the other analyses.

SD2 and SD4 seemed to be rather similar for their petrographic features. The presence of albite as an accessory mineral could suggest an affinity with Carrara, despite their textural properties not being strictly related to this basin, but resembling Pentelic marble instead. MGS and isotopic data indicate various sources, but their crossed interpretation is concordant on Carrara and Hymettus districts. However, generally, Hymettus marbles show a characteristic foliation [21] not observed in SD2 and SD4. Moreover, Hymettus is macroscopically different, being a bluish-grey marble [16] with characteristic greyish banding with a straight and parallel course, easily confused in the past with Proconnesian typology, as reported by [26] in the study of the Trajan's arch at Ancona (Italy). In addition, both SD2 and SD4 fell into the Carrara field in the Mn diagram.

Although the samples did not exhibit the typical microstructure of Carrara marble s.s. (e.g., homeoblastic structure, polygonal fabric, well visible triple points) all the other data

(Mn content, isotopic ratios, accessory minerals) suggest this as a plausible district of provenance. This supposed discrepancy could be explained with the variability observed—both macroscopically and, consequently, microscopically—in the marble outcropping quarried in the Apuan Alps district [27–34]. Moreover, different sites of excavation are not easy to distinguish, even with the available isotopic ratios as reported by [16,35,36] regarding the discrimination between the main Carrara quarries Torano, Colonnata and Miseglia.

The results obtained by MGS for SD3 indicated Carrara, Pentelicon, Docimium (Afyon), Hymettus or Göktepe as probable sources. Mn content and isotopic ratio allowed us to restrict this range to Docimium (Afyon) and Pentelicon. The textural features seemed nearer to the first one, especially for the strained crystals; in addition, dolomite, which is rather common in Pentelicon, and in subordinated amount to calcite [19], is instead rare or absent in Docimium [19]. In this regard, no dolomite was recognized by EPMA analysis in the SD3 sample.

SD5 and SD6 samples did not undergo all the methodologies, as clarified previously.

MGS suggested Carrara, Göktepe, Docimium (Afyon) and Hymettus as sources of SD5 and SD6. The textural and petrographic features seemed more like Carrara marble.

SD6 fell only into Carrara and Hymettus fields in the isotopic ratio diagram, excluding the other ones. Although the discrimination between Göktepe and Carrara is not easy and is still currently strongly debated [8,9,16,37,38], the Carrara hypothesis is also supported by the presence of accessory minerals such as firstly albite (a typical accessory of Carrara), and secondary apatite, muscovite and titanium oxide. Although they are common in many marbles, they can be all present in Carrara marble [19,24].

## 4. Conclusions

The archaeometric approach applied in the study of the "Madonna della Febbre" altar provided important information about the provenance of the marbles used in the sculptural group, contributing to reconstruct the historical context related to the San Domenico church.

The combination of complementary techniques allowed us to collect petrographic and geochemical data crucial for identifying the probable districts from which the marbles were quarried.

The results suggested that the elements of the altar were realized with different stones, some of them not of clear provenance. The trabeation (SD1 sample) showed unique features not observed in other portions of the sculptural complex: it consisted of a medium-grain-size marble with strongly deformed crystals. The data collected were opposing, and did not match clearly and coherently with a single district of provenance. The two Dominican saints at the sides of the Madonna, san Domenico (SD2 sample) and san Tommaso d'Aquino (SD4) were sculpted in the same material, presumably ascribable to the Carrara district. The frontal sample (SD3) showed elements similar to Docimium (Afyon) marble. Finally, the sculpture of the Madonna (SD5) and her pedestal (SD6) were clearly attributable to Carrara.

The uncertain provenance of some samples could be linked to the small size of the fragments analyzed providing only partial information, especially on the texture. The research conducted, nonetheless, represents an important starting point in the study of the materials, especially marbles, used in the architectural complex of the San Domenico church, and suggests the necessity of deepening subsequent investigations, for example, by increasing the areas of sampling and extending the archaeometric study to the other stone materials present in the church.

Overall, the variability of the materials used suggests the importance of the entire monument and highlights the complexity of its history. In this regard, many issues could be investigated: the use of different marbles in the statues and in the other decorative elements could contribute to defining the historical–artistic context of the sculptural group, and to identify the presence of one or more authors working in the same epoch or in subsequent phases. In addition, the results achieved could inform further hypotheses; on one hand, they could be helpful to reconstruct past trade routes and trends, e.g., justifying the choice of marble coming both from Italy (Carrara) and Asian Minor (e.g., Docimium). On the other

hand, these different materials could be related to the widespread practice of the re-use of ancient marble. Generally, the data collected did not allow us to confirm our hypothesis, but could contribute to future studies regarding the San Domenico complex.

**Supplementary Materials:** The following supporting information can be downloaded at: https://www.mdpi.com/article/10.3390/min12030284/s1, Table S1: Manganese concentration (in ppm) determined by ICP-MS in samples (SD2, SD3, SD4) and Standard Reference Material (Mica Schist SDC1). The measurements were repeated three times, thus mean and deviation standard were calculated and reported in the table.

**Author Contributions:** Conceptualization, D.B.; methodology, N.R. and D.B.; validation, N.R. and D.B.; formal analysis, N.R. and D.B.; investigation, N.R. and D.B.; resources, S.B.; data curation, N.R. and D.B.; writing—original draft preparation, N.R.; writing—review and editing, N.R. and D.B.; visualization, N.R.; supervision, D.B. All authors have read and agreed to the published version of the manuscript.

**Funding:** This research received no external funding.

**Data Availability Statement:** Data is contained within the article.

**Acknowledgments:** The authors would like to thank the anonymous reviewers for their suggestions during the revision of the manuscript.

**Conflicts of Interest:** The authors declare no conflict of interest.

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
