# Peer review of "Archaeometric Study of the White Marbles from “Madonna Della Febbre” Altar in San Domenico Church (Cosenza, Southern Italy)"

_minerals, doi:10.3390/min12030284_

Round 1
Reviewer 1 Report
Title: “Archaeometric study of the white marbles from Madonna della 2 Febbre altar in San Domenico Church (Cosenza-Southern Italy)”
Authors: Natalia Rovella, Stefania Bosco and Donatella Barca
Reviewer evaluation:
The paper reports about an interesting case-study and it recalls the recent discussion about the provenance determination for white marble. Enough care was also given to the description of the analytical protocol and results, but some details lack about the number of samples analysed by each technique and the motivation for the chosen protocol. The discussion can be definitely improved: additional insights may come from a more careful and comprehensive analysis of the results, as well as from the comparison with data in further references. Therefore, I recommend the paper acceptance after major revision.
Reviewer’s comments:
Specific comments:
INTRODUCTION
- Lines 47-48: I would provide a few more details here, as it might be meaningful to explain the different provenance of the marbles?
- Lines 79-81: would try to provide a few details on the most common exploitation sites in the 16th century and in Cosenza.
- Line 87: The determination of provenance for marbles is tricky on itself and several techniques are included in the list of possible protocols to reach the target. I would focus more on the state of the art of marble provenance protocols, rather than on different materials. I suggest to modify this paragraph and remove these references, which I consider not pertinent. This also helps the reader in the understanding of your methodology.
MATERIALS&METHODS
- Lines 95-99: Here I would also justify your choice in terms of sample location. Is there a reason why you limited the sampling to 6? Is each sample representative of a marble block that shows macro-features different than the others, like colour, grain structure, veins and foliations, state of preservation and the presence of impurities?
- Lines 113-115: Was the MGS estimated by a specific software or not? Please, provide further details.
- Line 124: were all or part of the samples analysed by IRMS? Please, provide details and motivate your choice.
- Line 143: 3 or 2 standard reference materials?
- Line 149: Please, motivate why ICP-MS has been applied to 3 samples only.
RESULTS
- Lines 157-160: this belongs to the methods, and it is in fact already stated at lines 110-112. Please, remove from here.
- Table 2: I would suggest replacing these abbreviations with the ones in the official mineral abbreviation list by D. L. Whitney and B. W. Evans.
- Figure 2: In my opinion, the caption can be shorter and details on each thin section can be provided in the results.
- Figure 3: I have the feeling that the resolution is not good for the plot from [26]. Please, provide higher quality plot. You could also consider a magnified view of it, since all your samples are below 5.
- Lines 245-252: this belongs to the Methods, please remove from here.
- Line 259: SD1 is distinct in MGS so based on both EPMA and POM, you could exclude some localities from this list. Please, give a comment in the discussion, as EPMA results are not fully considered there.
- Line 268: SD5 and SD6 show several common features: Plagioclase, mica, apatite and MGS. Based on it, can you shorten the list of possible localities? Please, include these findings in the discussion.
- Figure 4: Why is SD5 missing in the plot?
- Lines 282-286: How does this finding match with POM and EPMA analysis? Please, include in the discussion.
- Lines 282-286: What about SD5?
- Line 289: ased on this reference, Goktepe is excluded from the discussion, but Mn data are available and comparable for this element in previous references. I think it is worth discussing it, especially for SD2 and SD4: for them, Mn concentration is similar to Goktepe and from POM analysis this locality seems plausible.
DISCUSSION
- Lines 323-331: What about the EPMA and IRMS results for this sample? How can you explain apatite? Please, provide a full discussion for SD1.
- Line 337-338: These details (foliation, colour) are never mentioned in the results. Please, provide macroscopic details in the results.
- Line 347: is there any sample described in previous references that can match SD2 or SD4 results?
- Lines 351-356: it is not clear to me why you can exclude Pentelicon for SD3 from your results. Please, further motivate the identification.
CONCLUSIONS
- Line 381: Is Pentelicon not plausible?
- Lines 384-368: To what extent are your results compatible with supplies in 16th-century Cosenza? Could you provide any matching historical reference?
See details of comments in the pdf file of the manuscript “minerals-1604588-peer-review-v1_revised”.

Author Response
Reviewer evaluation:
The paper reports about an interesting case-study and it recalls the recent discussion about the provenance determination for white marble. Enough care was also given to the description of the analytical protocol and results, but some details lack about the number of samples analysed by each technique and the motivation for the chosen protocol. The discussion can be definitely improved: additional insights may come from a more careful and comprehensive analysis of the results, as well as from the comparison with data in further references. Therefore, I recommend the paper acceptance after major revision.
Reviewer’s comments:
Specific comments:
INTRODUCTION
Lines 47-48: I would provide a few more details here, as it might be meaningful to explain the different provenance of the marbles?
Unfortunately, the available information is lacking; there is no data linking the various authors to the probable choice of a particular type of marble.
Lines 79-81: would try to provide a few details on the most common exploitation sites in the 16th century and in Cosenza.
The historical sources about the exploitation sites of marble used in Cosenza are too fragmentary, so the authors preferred to not include them in the manuscript. In “Panariello, M.; Cura, M. Colligite fragmenta. Il lapidario di Cosenza fra storia, arte e restauro; ConSenso-Esperide Eds.: Briatico-Vibo Valentia, Italy, 2019; pp. 73-76. ISBN 978-88-97715-66-5” the authors cite some sources about the intention of some collaborators of the Cosenza Curia, during those years, to go in the Carrara areas to choose some marble blocks, but no other trace of this trade was found in the urban archives. Moreover, as reported in the text, the phenomenon of the re-using of ancient marble blocks was widespread in that period.
Line 87: The determination of provenance for marbles is tricky on itself and several techniques are included in the list of possible protocols to reach the target. I would focus more on the state of the art of marble provenance protocols, rather than on different materials. I suggest to modify this paragraph and remove these references, which I consider not pertinent. This also helps the reader in the understanding of your methodology.
In the previous lines, the authors talked about the issue of the provenance of the marbles and the protocols used in a typical archaeometric approach. In this part, the authors would like to underline the validity and the versatility of the same archaeometric approach and of the methodologies connected to it also in other ancient materials related to the world of cultural heritage for completeness of information. For these reasons, the authors would prefer not to change the sentence.
MATERIALS&METHODS
Lines 95-99: Here I would also justify your choice in terms of sample location. Is there a reason why you limited the sampling to 6? Is each sample representative of a marble block that shows macro-features different than the others, like colour, grain structure, veins and foliations, state of preservation and the presence of impurities?
The officer influenced decisively the number of samples to take, so the authors decided to select the areas most representative for every marble block composing the entire complex.
The following sentence was improved: “The sampling was carried out in not visible areas of the different marble blocks constituting the elements of the altar”.
Lines 113-115: Was the MGS estimated by a specific software or not? Please, provide further details.
The measurements were collected by the same AxioLab microscope software used for the acquisition of the images in thin section. This detail was added in the text.
Line 124: were all or part of the samples analysed by IRMS? Please, provide details and motivate your choice.
The sentence was added: SD5 was not analysed because of the insufficient amount of sample.
Line 143: 3 or 2 standard reference materials?
1, the authors corrected the mistake.
Line 149: Please, motivate why ICP-MS has been applied to 3 samples only.
The sentence was added: It was not possible to analyse SD1, SD5, SD6 because of the insufficient amount of available sample.
RESULTS
Lines 157-160: this belongs to the methods, and it is in fact already stated at lines 110-112. Please, remove from here.
The authors removed the lines as suggested.
Table 2: I would suggest replacing these abbreviations with the ones in the official mineral abbreviation list by D. L. Whitney and B. W. Evans.
The authors corrected the abbreviations.
Figure 2: In my opinion, the caption can be shorter and details on each thin section can be provided in the results.
The caption was shortened.
Figure 3: I have the feeling that the resolution is not good for the plot from [26]. Please, provide higher quality plot. You could also consider a magnified view of it, since all your samples are below 5.
The authors checked the resolution and probably it is linked with the pdf conversion. The authors prefer to maintain the original plot.
Lines 245-252: this belongs to the Methods, please remove from here.
The authors moved this part in to the Methods.
Line 259: SD1 is distinct in MGS so based on both EPMA and POM, you could exclude some localities from this list. Please, give a comment in the discussion, as EPMA results are not fully considered there.
The authors added the sentence in the discussion about SD1: “Finally, small apatites <40 µm identified by EPMA-EDS could suggest Paros provenance, but this thesis is not supported by the other analyses.”
Line 268: SD5 and SD6 show several common features: Plagioclase, mica, apatite and MGS. Based on it, can you shorten the list of possible localities? Please, include these findings in the discussion.
This line is related only to the comment about the localities where titanium oxides are present as accessories according to literature. The provenance of SD5 and SD6 is deepened in the discussion, where Carrara is identified as the source.
Figure 4: Why is SD5 missing in the plot?
The authors clarified in the methods that “SD5 was not analysed because of the insufficient amount of sample.”
Lines 282-286: How does this finding match with POM and EPMA analysis? Please, include in the discussion.
The results about this sample are not so unanimous to indicate a clear source and this issue was dealt with the discussion.
Lines 282-286: What about SD5?
As reported before, the authors clarified in the methods that “SD5 was not analysed because of the insufficient amount of sample.”
Line 289: based on this reference, Goktepe is excluded from the discussion, but Mn data are available and comparable for this element in previous references. I think it is worth discussing it, especially for SD2 and SD4: for them, Mn concentration is similar to Goktepe and from POM analysis this locality seems plausible.
The authors as reported in the discussion, excluded Goktepe because albite as accessory and IRMS results are not compatible with this source.
DISCUSSION
Lines 323-331: What about the EPMA and IRMS results for this sample? How can you explain apatite? Please, provide a full discussion for SD1.
The results about this sample are not so unanimous to indicate a clear source and this issue was dealt with and deepened in the discussion.
Line 337-338: These details (foliation, colour) are never mentioned in the results. Please, provide macroscopic details in the results.
The authors clarified it in the description of the marble complex in the introduction: “it is constituted by whitish marble blocks without particular foliations”.
Line 347: is there any sample described in previous references that can match SD2 or SD4 results?
Not clearly, in fact, the references cited highlight the discrepancy of many marbles that do not show the “classical” Carrara aspect also if they come anyway from the same Apuan area and are geochemically comparable.
Lines 351-356: it is not clear to me why you can exclude Pentelicon for SD3 from your results. Please, further motivate the identification.
The authors considering the textural features and the total absence of dolomite indicate Docimium as most probable source and exclude Pentelicon.
CONCLUSIONS
Line 381: Is Pentelicon not plausible?
The authors believe the provenance from Docimium is most probable than Pentelicon based on the available data
Lines 384-368: To what extent are your results compatible with supplies in 16th-century Cosenza? Could you provide any matching historical reference?
They could be compatible, because there was a large re-use of ancient marbles and also the marbles from Carrara district were widespread in Italy; however as reported in the review about lines 79-81, the historical sources about the exploitation sites of marble used in Cosenza are too fragmentary, so the authors preferred to not include them in the manuscript.

Reviewer 2 Report
This is an interesting manuscript about the provenance analysis of marbles from the Madonna della Febbre altar.
Major changes
line 153: certified reference material results should be given in full in the supplementary info so it is possible to check the quality of the analysis
Fig. 5: This graph has a logarithmic scale on the x-axis. This means that the lines for SD2 and SD3 are drawn in the wrong position (they should be further right). This also means that for SD3, there is no overlapping source area in the image (needs to be fixed in line 290 as well). In this context it becomes clear that Table 2 should also list the standard deviation/uncertainty for the Mn results. This result suggests the origin of Docimium for SD3 is not that likely, assuming that the sample is representative of the whole, and that the literature data accurately represent the potential source materials.
Minor changes
line 94: how were these marble fragments taken? With a saw? 1cm size – does this refer to 1 cm3?
line 124 and 274: the delta symbols are missing (δ13C, δ18O). 13C and 18O by themselves are not isotope ratios, they are only isotopes.
line 135: how was the sample broken down and then powdered?
line 143: mentions three standard reference materials, but only two reference materials are named
Several language issues are present, the first few of which are listed below (but this list is far from complete).
line 31: “author” is this supposed to refer to the architect? also line 31 “little clear” à unclear
line 77: “used since past” change to e.g. “in the past”
line 85 “It will be used different methodologies..” à “Different methodologies were used”
also in line 315: “Diagram edited by [25]” means that the people from [25] edited your diagram, presumably you mean “Diagram by [25], edited to include Mn concentrations from this study” or similar
Author Response
Reviewer 2
This is an interesting manuscript about the provenance analysis of marbles from the Madonna della Febbre altar.
Major changes
line 153: certified reference material results should be given in full in the supplementary info so it is possible to check the quality of the analysis.
The results about certified reference material were uploaded in the supplementary file.
Fig. 5: This graph has a logarithmic scale on the x-axis. This means that the lines for SD2 and SD3 are drawn in the wrong position (they should be further right). This also means that for SD3, there is no overlapping source area in the image (needs to be fixed in line 290 as well). In this context it becomes clear that Table 2 should also list the standard deviation/uncertainty for the Mn results. This result suggests the origin of Docimium for SD3 is not that likely, assuming that the sample is representative of the whole, and that the literature data accurately represent the potential source materials.
The data about Mn, deviation standard, accuracy was uploaded in the supplementary file.
Minor changes
line 94: how were these marble fragments taken? With a saw? 1cm size – does this refer to 1 cm3?
By scalpel, the authors added it. Yes, 1 cm3
line 124 and 274: the delta symbols are missing (δ13C, δ18O). 13C and 18O by themselves are not isotope ratios, they are only isotopes.
The authors added the symbols.
line 135: how was the sample broken down and then powdered?
By agate mortar, the authors added it.
line 143: mentions three standard reference materials, but only two reference materials are named
The authors corrected the mistake, the standard material is one.
Several language issues are present, the first few of which are listed below (but this list is far from complete).
line 31: “author” is this supposed to refer to the architect? also line 31 “little clear” à unclear
Yes, it is referred to the architect (the authors substituted it). Little clear was changed in unclear.
line 77: “used since past” change to e.g. “in the past”
The authors corrected it.
line 85 “It will be used different methodologies..” à “Different methodologies were used”
The authors corrected it.
also in line 315: “Diagram edited by [25]” means that the people from [25] edited your diagram, presumably you mean “Diagram by [25], edited to include Mn concentrations from this study” or similar
The authors corrected it.

Round 2
Reviewer 1 Report
After the first revision, the experimental and result sections gained completeness, but some minor errors are still present.
I find that the introduction still lacks some details, especially for what concerns the state of the art. Some suggestions had been provided and are again reported in the new revision for help (see pdf file 'minerals-1604588-peer-review-v2_revised').
Finally, I still consider some references [19-23] unnecessary self-citations and I strongly recommend to eliminate them. Please, include the ones related to marble analysis only.
I recommend the publication of the paper after minor revision.

Author Response
The author reports below the comments in italics to the Reviewer suggestions, thanking him/her for the precious support.
I find that the introduction still lacks some details, especially for what concerns the state of the art. Some suggestions had been provided and are again reported in the new revision for help (see pdf file 'minerals-1604588-peer-review-v2_revised').
The authors included the suggestions in the text as requested and indicated in the comments in the pdf file.
Finally, I still consider some references [19-23] unnecessary self-citations and I strongly recommend to eliminate them. Please, include the ones related to marble analysis only.
The authors modified the references as requested.